# Attribution of Predictive Uncertainties in Classification Models

**Iker Perez**[1]     **Piotr Skalski**[1]     **Alec Barns-Graham**[1]     **Jason Wong**[1]     **David Sutton**[1]

[1]Featurespace Research, Cambridge, United Kingdom

## Abstract

Predictive uncertainties in classification tasks are often a consequence of model inadequacy or insufficient training data. In popular applications, such as image processing, we are often required to scrutinise these uncertainties by meaningfully attributing them to input features. This helps to improve interpretability assessments. However, there exist few effective frameworks for this purpose. Vanilla forms of popular methods for the provision of saliency masks, such as SHAP or integrated gradients, adapt poorly to target measures of uncertainty. Thus, state-of-the-art tools instead proceed by creating *counterfactual* or *adversarial* feature vectors, and assign attributions by direct comparison to original images. In this paper, we present a novel framework that combines path integrals, counterfactual explanations and generative models, in order to procure attributions that contain few observable artefacts or noise. We evidence that this outperforms existing alternatives through quantitative evaluations with popular benchmarking methods and data sets of varying complexity.

## 1 INTRODUCTION

Model uncertainties often manifest aspects of a system or data generating process that are not exactly understood [Hüllermeier and Waegeman, 2021], such as the influence of model inadequacy or a lack of diverse and representative data used during training. The ability to quantify and attribute such uncertainties to their sources can help scrutinize aspects in the functioning of a predictive model, and facilitate interpretability or fairness assessments in important machine learning applications [Awasthi et al., 2021]. The process is especially relevant in Bayesian inferential settings, which find applications in domains such as natural language processing [Xiao and Wang, 2019], network analysis [Perez and Casale, 2021] or image processing [Kendall and Gal, 2017], to name only a few.

Thus, there exists a growing interest in methods for uncertainty estimation [e.g. Depeweg et al., 2018, Smith and Gal, 2018, Van Amersfoort et al., 2020, Tuna et al., 2021] for purposes such as procuring adversarial examples, active learning or *out-of-distribution* detection. Recent work has proposed mechanisms for the attribution of predictive uncertainties to input features, such as pixels in an image [Van Looveren and Klaise, 2019, Antoran et al., 2021, Schut et al., 2021], with the goal of complementing interpretability tools disproportionately centred on explaining model scores, and to improve transparency in deployments of predictive models. These methods proceed by identifying *counterfactual* (in-distribution) or *adversarial* (out-of-distribution) explanations, i.e. small variations in the value of input features which output new model scores with minimal uncertainty. This has helped understand the strengths and weaknesses of various models. However, the relative contribution of individual pixels to poor model performance is up to human guesswork, or assigned by plain comparisons between an image and its altered representation. We report that uncertainty attributions derived following these approaches perform poorly, when measured by popular quantitative evaluations of image saliency maps.

In this paper, our goal is to similarly map uncertainties in classification tasks to their origin in images, and to measure the relative contribution of each individual pixel. We show that popular attribution methods based on *segmentation* [Ribeiro et al., 2016], *resampling* [Lundberg and Lee, 2017a] or *path integrals* [Sundararajan et al., 2017] are easily re-purposed for this purpose. However, we evidence that naive applications of these approaches perform poorly. Thus, we present a new framework through a novel combination of path integrals, counterfactual explanations and generative models. Our approach is to attribute uncertainties by traversing a domain of integration defined in latent space, which connects a counterfactual explanation with its original im-

*Accepted for the 38th Conference on Uncertainty in Artificial Intelligence* (UAI 2022).

age. The integration is projected into the observable pixel space through a generative model, and starts at a reference point which bears no predictive uncertainty. Hence, *completeness* is satisfied and uncertainties are fully explained and decomposed over pixels in an image.

We note that relying on generative models has recently gained traction for interpretability and score attribution purposes [Lang et al., 2021]. Through our method, we show how to leverage these models in order to procure clustered saliency maps, which reduce the observable noise in vanilla approaches. Applied to uncertainty attribution tasks, the proposed approach outperforms vanilla adaptations of popular interpretability tools such as LIME [Ribeiro et al., 2016], SHAP [Lundberg and Lee, 2017a] or integrated gradients [Sundararajan et al., 2017], as well as *blur* and *guided* variants [Xu et al., 2020, Kapishnikov et al., 2021]. We further combine these methods with *Xrai* [Kapishnikov et al., 2019], a popular segmentation and attribution approach. The assessment[1] is carried out through both quantitative and qualitative evaluations, using popular benchmarking methods and data sets of varying complexity.

## 2  UNCERTAINTY ATTRIBUTIONS

Consider a classification task with a classifier $f : \mathbb{R}^n \times \mathcal{W} \rightarrow \Delta^{|\mathcal{C}|-1}$ of a fixed architecture. The weights $\boldsymbol{w} \in \mathcal{W}$ are presumed to be fitted to some available train data set $\mathcal{D} = \{\boldsymbol{x}_i, c_i\}_{i=1,2,\dots}$. Thus, the function $f(\boldsymbol{x}) \equiv f(\boldsymbol{x}, \boldsymbol{w})$ maps feature vectors $\boldsymbol{x} \in \mathbb{R}^n$ to an element in the standard $(|\mathcal{C}|-1)$-simplex, which represents membership probabilities across classes in a set $\mathcal{C}$. In the following, we are concerned with the *entropy* as a measure of predictive uncertainty, i.e.

$$H(\boldsymbol{x}) = -\sum_{c \in \mathcal{C}} f_c(\boldsymbol{x}) \cdot \log f_c(\boldsymbol{x}) \tag{1}$$

where $f_c(\boldsymbol{x})$ represents the predicted probability of class-$c$ membership. In Bayesian settings, we often consider a posterior distribution $\pi(\boldsymbol{w}|\mathcal{D})$ over weights in the model, and the term (1) may further be decomposed into *aleatoric* and *epistemic* components [Kendall and Gal, 2017]. These represent different types of uncertainties, including inadequate data and inappropriate modelling choices. For simplicity in the presentation, we defer those details to Section 1 in the supplementary material.

Popular *resampling* or *gradient*-based methods can easily be adapted in order to attribute measures of uncertainty such as $H(\boldsymbol{x})$ to input features in an image. This includes tools such as LIME [Ribeiro et al., 2016], SHAP [Lundberg and Lee, 2017a] or *integrated gradients* (IG) [Sundararajan et al., 2017]. In Figure 1, we show an example application of integrated gradients to *dogs versus cats* data (further examples

Original image

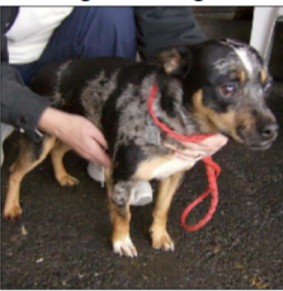

Entropy attributions

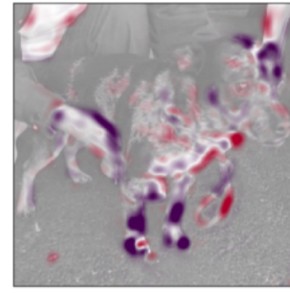

Aleatoric attributions

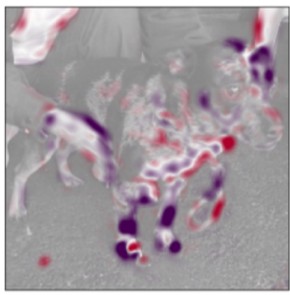

Epistemic attributions

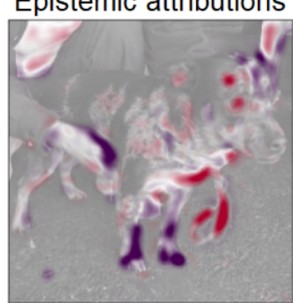

Figure 1: Example uncertainty attributions using integrated gradients. Classification task in *dogs versus cats* data. In red, positive attributions which *increase* entropy; in purple, negative attributions that *decrease* entropy.

are found in Section 3 in the supplementary material). In the figure, the regions in red are identified as contributors to predictive uncertainties. We readily comprehend why the model struggles to predict any single class, by observing that a leash and a human hand are problematic. To the best of our knowledge, no research has yet explored the possibility of using these attribution methods to identify sources of uncertainty. Nevertheless, quantitative evaluations presented in Section 4 show that this approach offers generally poor performance.

### 2.1  PATH INTEGRALS

For later reference, we illustrate the above uncertainty attribution procedure with integrated gradients. In primitive form, a path method explains a scalar output $F(\boldsymbol{x})$ using a *fiducial* image $\boldsymbol{x}^0$ as reference, which is presumably not associated with any class observed in training data. The importance attributed to *pixel $i$* for the purposes of explaining the quantity $F(\boldsymbol{x})$ is given by

$$\text{attr}_i^{\delta}(\boldsymbol{x}) = \int_0^1 \frac{\partial F(\delta(\alpha))}{\partial \delta_i(\alpha)} \frac{\partial \delta_i(\alpha)}{\partial \alpha} d\alpha$$

where $\delta : [0,1] \rightarrow \mathbb{R}^n$ represents a curve with endpoints at $\delta(0) = \boldsymbol{x}^0$ and $\delta(1) = \boldsymbol{x}$. Here, $\sum_i \text{attr}_i(\boldsymbol{x}) = F(\boldsymbol{x}) - F(\boldsymbol{x}^0)$ follows from the *gradient theorem* for line integrals, s.t. the difference in output values decomposes over the

sum of attributions. Commonly, $F(\boldsymbol{x}) = f_c(\boldsymbol{x})$ represents the classification score for a class $c \in \mathcal{C}$ s.t. attributions capture elements in an image that are associated with this class. In order to attribute uncertainties, we readily assign $F(\boldsymbol{x}) = H(\boldsymbol{x})$, and thus combine scores across all classes with aims to identify pixels that *confuse* the model.

**Integrated Gradients**. Here, $\delta$ is parametrised as a straight path between a fiducial and the observed image, i.e. $\delta(\alpha) = \boldsymbol{x}^0 + \alpha(\boldsymbol{x} - \boldsymbol{x}^0)$, and the above simplifies to

$$\text{IG}_i(\boldsymbol{x}) = (x_i - x_i^0) \times \int_0^1 \frac{\partial H(\boldsymbol{x}^0 + \alpha(\boldsymbol{x} - \boldsymbol{x}^0))}{\partial x_i} d\alpha,$$

which corresponds to entropy attributions in Figure 1 ( see Section 1 in the supplementary material for its decomposition into aleatoric and epistemic attributions).

Integrated gradients offers an efficient approach to produce attributions with differentiable models, as an alternative to layer-wise relevance propagation [Montavon et al., 2019] or DeepLift [Shrikumar et al., 2017], and there exist several adaptations and extensions [Smilkov et al., 2017, Xu et al., 2020, Kapishnikov et al., 2021]. However, attributions are heavily influenced by differences in pixel values between $\boldsymbol{x}$ and $\boldsymbol{x}^0$, and the fiducial choice defaults to a black (or white) background. This fails to attribute importances to black (or white) pixels and is considered problematic [Sundararajan et al., 2017], leading to proposed *blurred* or *black+white* alternatives [Lundberg and Lee, 2017b, Kapishnikov et al., 2019]. Additionally, $\delta$ transitions the path $\boldsymbol{x}^0 \rightsquigarrow \boldsymbol{x}$ *out-of-distribution* [Jha et al., 2020, Adebayo et al., 2020], i.e. through intermediary images not representative of training data, leading to noise and artefacts in attributions.

## 3 METHODOLOGY

We describe the proposed method for uncertainty attributions summarised in Algorithm 1. This combines path integrals with a generative process to define a domain of integration. We use a counterfactual fiducial bearing no relation to causal inference [Pearl, 2010], i.e. an alternative *in distribution* image $\boldsymbol{x}^0$ similar to $\boldsymbol{x}$ according to a suitable metric, s.t. $f(\boldsymbol{x}^0)$ bears close to 0 predictive uncertainty.

We choose to leverage a *variational auto-encoder* (VAE) as the generative model. As customary, this is composed of a unit-Gaussian data-generating process of arbitrary dimensionality $m \ll n$, along with an image decoder $\psi : \mathbb{R}^m \to \mathbb{R}^n$. Here, $\boldsymbol{z}|\boldsymbol{x} \sim \mathcal{N}(\phi_\mu(\boldsymbol{x}), \phi_\sigma(\boldsymbol{x}))$ represents the approximate posterior in latent space, with mean and variance encoding functions $\phi_\mu, \phi_\sigma : \mathbb{R}^n \to \mathbb{R}^m$.

### 3.1 DOMAIN OF INTEGRATION

The domain of integration is defined as a curve across endpoints $\boldsymbol{x}^0 \rightsquigarrow \boldsymbol{x}$. We select the fiducial as a decoded image

$\boldsymbol{x}^0 = \psi(\boldsymbol{z}^0)$, where $\boldsymbol{z}^0$ is the solution to the constrained optimization problem

$$\underset{\boldsymbol{z} \in \mathbb{R}^m}{\arg \min} \left[ d(\psi(\boldsymbol{z}), \boldsymbol{x}) + \frac{1}{2m} \sum_j z_j^2 \right] \qquad (2)$$

$$\text{subject to} \quad \|e_{\hat{c}} - f(\psi(\boldsymbol{z}))\| < \varepsilon$$

for an infinitesimal $\epsilon > 0$. Here, $\hat{c} = \arg \max_i f_i(\boldsymbol{x})$ is the predicted class by the classifier, and $e_i$ is the unit indicator vector at index $i$. The metric $d(\cdot, \cdot)$ may be chosen to be the cross-entropy or mean absolute difference over pixel values in an image. The right-most term is the negative log-density (up to proportionality) of $\boldsymbol{z}$ in a latent space of dimensionality $m > 0$; this restricts the search *in-distribution* and ensures robustness to overparametrisation of the latent space within our experiments.

Hence, we retrieve a counterfactual fiducial which (i) is classified in the same class as $\boldsymbol{x}$ and (ii) bears close to zero predictive uncertainty. In practice, we approximate (2) through the penalty method, i.e. an unconstrained search with a large penalty on

$$d_{\mathcal{X}}(e_{\hat{c}}, f(\psi(\boldsymbol{z}))) = -\log f_{\hat{c}}(\psi(\boldsymbol{z})),$$

i.e. the cross-entropy between the predicted class $\hat{c}$ and the membership vector $f(\psi(\boldsymbol{z}))$ given a decoding $\psi(\boldsymbol{z})$. We proceed by gradient descent initialised at $\phi_\mu(\boldsymbol{x})$, the encoder's mean.

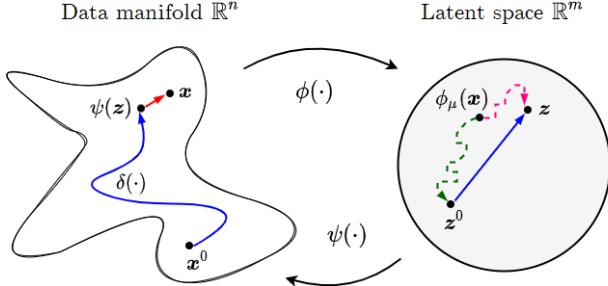

Figure 2: Procedural sketch to generate a path of integration. Here, *fiducial $\boldsymbol{z}^0$* and *reconstruction $\boldsymbol{z}$* points are optimized in latent space by gradient descent, starting initially from the encoding of $\boldsymbol{x}$ (dashed lines). A connecting straight path (in blue) is projected to the data-manifold and augmented with an interpolating component (in red).

**Integration Path**. We further leverage the decoder as a generative process to parametrise a curve $\delta_\psi : [0, 1] \to \mathbb{R}^n$, by following the steps displayed in Figure 2, s.t. $\delta_\psi(\alpha) = \psi(\boldsymbol{z}^0 + \alpha(\boldsymbol{z} - \boldsymbol{z}^0))$ where

$$\boldsymbol{z} = \underset{\boldsymbol{z} \in \mathbb{R}^m}{\arg \min} \left[ d(\psi(\boldsymbol{z}), \boldsymbol{x}) + \frac{1}{2m} \sum_j z_j^2 \right]$$

is also optimised by gradient descent initialised at $\phi_\mu(\boldsymbol{x})$. This is an unconstrained optimisation problem analogue

**Algorithm 1:** Generative Uncertainty Attributions

---

**input** : Feature vector $\boldsymbol{x}$, predictive distribution $f(\cdot)$ and distance metric $d(\cdot,\cdot)$.
   VAE encoder $\phi(\cdot)$ and decoder $\psi(\cdot)$, penalty $\lambda >> 0$ and learning rate $\nu > 0$.

**output** : Attributions $\mathrm{attr}_i^{\delta_\psi}(\boldsymbol{x})$, $i = 1, \dots, n$.

Initialise $\boldsymbol{z}^0 = \boldsymbol{z} = \phi_\mu(\boldsymbol{x})$;

Compute predicted class $\hat{c} = \arg\max_i f_i(\boldsymbol{x})$;

**while** $\mathcal{L}_1$ *not converged* **do**

$$\mathcal{L}_1 \leftarrow d(\psi(\boldsymbol{z}^0), \boldsymbol{x}) + \frac{1}{2m}\sum_j z_j^2 - \lambda \log f_{\hat{c}}(\psi(\boldsymbol{z})) \quad \text{and} \quad \boldsymbol{z}^0 \leftarrow \boldsymbol{z}^0 - \nu\nabla_{\boldsymbol{z}}\mathcal{L}_1$$

**end**

**while** $\mathcal{L}_2$ *not converged* **do**

$$\mathcal{L}_2 \leftarrow d(\psi(\boldsymbol{z}), \boldsymbol{x}) + \frac{1}{2m}\sum_j z_j^2 \quad \text{and} \quad \boldsymbol{z} \leftarrow \boldsymbol{z} - \nu\nabla_{\boldsymbol{z}}\mathcal{L}_2$$

**end**

Approximate $\mathrm{attr}_i^{\delta_\psi}(\boldsymbol{x})$, $i = 1, \dots, n$ in (3) along $\delta_{\psi, \boldsymbol{z}^0 \to \boldsymbol{z}}$ through trapezoidal integration.

---

to (2). Consequently, the path $\delta_\psi$ offers trajectory between a counterfactual $\delta_\psi(0) = \psi(\boldsymbol{z}^0) = \boldsymbol{x}^0$ and a reconstruction $\delta_\psi(1) = \psi(\boldsymbol{z})$ of the image $\boldsymbol{x}$. In order to correct for mild reconstruction errors, we finally augment the domain of integration through a *vanilla* straight path between the end-points $\psi(\boldsymbol{z}) \rightsquigarrow \boldsymbol{x}$. We display a few examples of this procedure on MNIST digits within Figure 3. Overall, the difference in predictive entropy or model scores between a reconstruction $\psi(\boldsymbol{z})$ and its original counterpart $\boldsymbol{x}$ are not observed to be significant within our experiments.

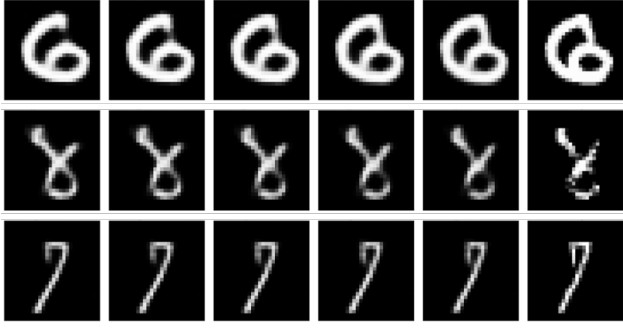

Figure 3: An example of *in-distribution* curves connecting fiducial (left-most) and real (right-most) data points, on MNIST digits data. Digits on the left bear no predictive uncertainty in classification.

### 3.2 LINE INTEGRAL FOR ATTRIBUTIONS

For simplicity, we restrict the formulae to the *in-distribution* component along the curve $\delta_\psi : [0,1] \to \mathbb{R}^n$ defined in Subsection 3.1, and we ignore the straight path connecting $\psi(\boldsymbol{z}) \rightsquigarrow \boldsymbol{x}$. We require the total differential of the entropy $H(\cdot)$ wrt $\boldsymbol{z}$ in latent space; however, we wish to retrieve importances for features $\boldsymbol{x}$ in the original data manifold

within $\mathbb{R}^n$. To this end, the attribution at index $i = 1, \dots, n$ is given by

$$\mathrm{attr}_i^{\delta_\psi}(\boldsymbol{x}) = \sum_{j=1}^m (z_j - z_j^0) \int_0^1 \frac{\partial H(\delta_\psi(\alpha))}{\partial \delta_{\psi,i}(\alpha)} \frac{\partial \delta_{\psi,i}(\alpha)}{\partial z_j} d\alpha. \tag{3}$$

Intuitively, we compute the total derivative of $H(\cdot)$ wrt $\alpha$ in the integration path, using the chain rule. We decompose the calculation over indices in pixel space, and further undertake summation over contributions in latent space. In Figure 4, we show an example that compares attributions in (3) versus vanilla integrated gradients. There, we find a *CelebA* image [Liu et al., 2015] with tags for the presence of a *smile*, *arched eyebrows* and *no bags under the eyes*.

### 3.3 PROPERTIES

Due to *path independence* and noting that $H(\boldsymbol{x}^0) \approx 0$ by definition, importances drawn from a trajectory $\delta_\psi(\cdot)$ as parametrised in Subsection 3.1 will approximately account for **all** of the uncertainty in a posterior predictive task, i.e.

$$H(\boldsymbol{x}) \approx \int_0^1 \nabla H(\delta_\psi(\alpha)) d\alpha = \sum_{i=1}^n \mathrm{attr}_i^{\delta_\psi}(\boldsymbol{x}),$$

and this is commonly referred to as *completeness*. Additionally, the reliance on path derivatives along with the rules of composite functions ensure that both fundamental axioms of *sensitivity(b)* (i.e. *dummy property*) along with *implementation invariance* are inherited, and we refer the reader to Friedman [2004], Sundararajan et al. [2017] for the technical details. Importantly, the attribution will be zero for any index which does not influence the classifier.

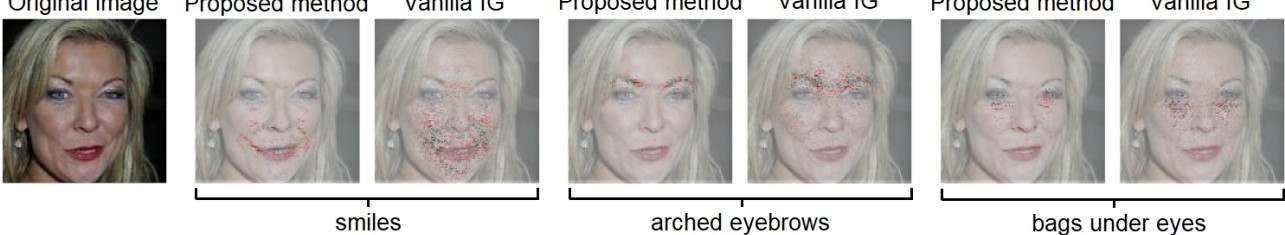

Figure 4: Comparison of uncertainty attributions on a *CelebA* image. We compare attributions for three classifiers, which measure the presence (or lack) of *smiles* (left), *arched eyebrows* (centre), and *bags under eyes* (right). Red pixels contribute by increasing uncertainties, in green we find contributions towards decreasing uncertainties.

### 3.3.1 The Role of the Autoencoder

A VAE is arguably not the best generative model for reconstructing sharp images with high fidelity. However, it is stable during training and efficient in sampling, furthermore, the encoder provides a mean to efficiently select starting values $\phi_\mu(\boldsymbol{x})$ during latent optimisation tasks [cf. Antoran et al., 2021]. In Section 2 within the supplementary material we offer a robustness assessment of our results to variations in the autoencoder, and we report on negligible changes in performance. We achieve consistency even in large over-parametrised latent spaces, due to Gaussian priors in the optimisation procedures in Subsection 3.1, which define the integration path.

Alternative models can be used to define integration paths. *Generative adversarial networks* have gained relevance as a means to facilitate interpretability in classification tasks [Lang et al., 2021], however, training can be unstable and identifying counterfactual references is infeasible. This also presents a problem with *autoregressive models* [Van den Oord et al., 2016], which are further inefficient in sampling and would pose long optimisation times in latent space.

### 3.3.2 Non-Generative Integration Paths

For simplicity, a counterfactual fiducial image $\boldsymbol{x}^0 = \psi(\boldsymbol{z}^0)$ as described in (2) can also be combined with a straight or *guided* [Kapishnikov et al., 2021] integration path $\psi(\boldsymbol{z}^0) \rightsquigarrow \boldsymbol{x}$. In application to simple grey-scale images, this path is unlikely to transverse *out-of-distribution* due to the proximity between a fiducial and the original image $\boldsymbol{x}$. In our experiments, we test these variants and report that they fare relatively well in explainability tasks with simple images; however, their performance degrades on complex RGB pictures involving facial features.

## 4 EXPERIMENTS

Uncertainty attributions are commonly facilitated through generative and adversarial models, and can thus be computationally expensive to produce. Consequently, they have

traditionally only been evaluated on simple data sets [cf. Antoran et al., 2021, Schut et al., 2021]. Here, we similarly apply our proposed methodology to classification models in the image repositories *MNIST handwritten digits* [LeCun and Cortes, 2010] and *fashion-MNIST* [Xiao et al., 2017]. However, we also extend evaluation tasks to high resolution facial images in *CelebA* [Liu et al., 2015].

We evaluate the performance both quantitatively and qualitatively, and we compare the results to path methods including *vanilla* integrated gradients [Sundararajan et al., 2017], as well as *blur* and *guided* variants [Xu et al., 2020, Kapishnikov et al., 2021]. We test these approaches with *plain*, *black+white* (B+W) and *counterfactual* fiducials, and we combine the saliency maps with Xrai [Kapishnikov et al., 2019], a popular segmentation and attribution approach. We also evaluate pure counterfactual approaches for uncertainty attributions, which assign importances by directly comparing pixel values between an image and its counterfactual. For this, we include most recent *CLUE* attributions [Antoran et al., 2021] in the assessment. For completeness, we finally add adaptations of *LIME* [Ribeiro et al., 2016] and *kernelSHAP* [Lundberg and Lee, 2017a]. Implementation details are found in the supplementary Section 4. Source code for reproducing results can be found at github.com/Featurespace/uncertainty-attribution.

### 4.1 PERFORMANCE METRICS

In order to produce quantitative evaluations we resort to *smallest sufficient region* methods popularised in recent literature [see Petsiuk et al., 2018, Kapishnikov et al., 2019, Covert et al., 2020, Lundberg et al., 2020], which evaluate the quality of saliency maps in the absence of ground truths. These are suitable for our repeated assessments over multiple methods and data sets, as they do not require for specialised model retrains [cf. Hooker et al., 2019, Jethani et al., 2021]. The methods proceed by revealing pixels from a masked image, in order of importance as determined by attribution values, and changes in classification scores, predictive entropy or image information content are monitored. Alternatively, the process may be carried backwards by re-

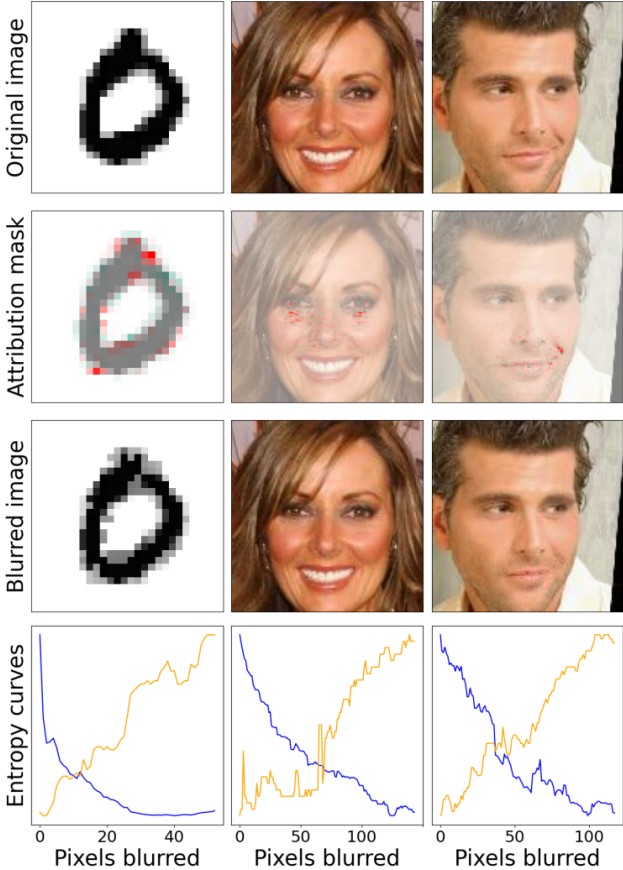

Figure 5: Normalised variation in predictive entropy (decreasing, blue) and image information content (increasing, orange) as pixels most contributing to uncertainty are sequentially blurred. Classification task on digits (left), bags under eyes (centre) and smiles (right). Information content approximated by compressed file sizes.

moving or resampling pixels from the original image, and we show an example of this process in Figure 5. We use blurring as a masking mechanism [cf. Kapishnikov et al., 2019], since other alternatives lead to masked images significantly out of distribution, i.e. non representative of training data. We evaluate two inclusion and removal metrics suitable to measure changes in predictive uncertainty.

**Inclusion Methods.** We measure the *entropy information curve* (EIC) in a manner analogue to *performance information curves* (PICs) discussed in Kapishnikov et al. [2019, 2021]. For an image $x$ with $n$ pixels, we define a sequence $\{x^i\}_{i=0,...,n}$ that transitions from a blurred reference $x^0 = x_{blurred}$ towards $x^n = x$, by revealing pixels in order of contribution to decreasing the entropy. We evaluate

$$EIC_i = \frac{1}{|\mathcal{X}|} \sum_{x \in \mathcal{X}} \frac{H(x^i)}{H(x_{blurred})}$$

across indexes in the transition $x_{blurred} \xrightarrow{i=1,...,n} x$, which

retrieves an average over images in each data set $\mathcal{X}$ (in the presence of significant outliers, we report on median values). The EIC measures the variation in overall predictive entropy and can be computed on unlabelled data. It is assessed versus the information content in the images as pixels are revealed Kapishnikov et al. [2019, 2021], which can be approximated by file sizes or the second order Shannon entropy.

**Best Removal Methods.** We measure *uncertainty reduction curves*, i.e. the relative uncertainty that an attribution method can remove from an image $x$. We use the inverse sequence $\{x^i\}_{i=0,...,n}$, which transitions from $x^0 = x$ towards a *blurred* image $x^n = x_{blurred}$. We evaluate

$$URC_i = \frac{1}{|\mathcal{X}|} \sum_{x \in \mathcal{X}} max_{r \leq i} \left[ 1 - \frac{H(x^r)}{H(x)} \right],$$

i.e. the best percentage reduction in predictive uncertainty that can be explained away by *blurring* up to $i$ pixels, in decreasing order of contribution to uncertainty.

### 4.2 QUANTITATIVE EVALUATION

In Table 1 we report on (i) the area over the entropy information curve and (ii) percentile points in the uncertainty reduction curve, for the various attribution methods and data sets analysed in this paper. We explore 5 classification tasks, including the presence of *smiles*, *arched eyebrows* and *eye-bags* in CelebA images. In all cases, high values represent better estimated performance. The metrics are evaluated on images that were excluded during model training. Attribution methods have been implemented with default parameters, where available, and we offer details in the supplementary Section 4. Blurring is performed with a Gaussian kernel, and the standard deviation is tuned individually for each classification task. We choose the minimum standard deviation s.t. a model's predictive uncertainty for the fully blurred images is maximised. *KernelSHAP* evaluations are offered only for data sets with small resolution images, due to the computational complexity associated with undertaking the recommended amount of image perturbations.

The results show that a generative method as presented in this paper is better suited to explain variations in predictive entropy, as well as explaining away sources of uncertainty. Results suggest that improvements over the explored alternatives are of significance in classification tasks with high resolution images concerning facial features. In application to low resolution grey scale images, the results also show that popular attribution approaches, such as integrated gradients, guided integrated gradients or SHAP require a counterfactual fiducial to perform well, which must still be produced through a generative model. In these cases, good performance is a consequence of low dissimilarity between an image and its baseline (see Subsection 3.3.2), s.t. simple integration paths remain in-distribution.

Table 1: Area over the entropy information curve and percentile points in uncertainty reduction curves, across attribution methods and classification tasks. Metrics procured wrt approximated and normalised information content of images.

| Method | Area over Entropy Information Curve | | | | | Uncertainty Reduction Curve | | | | | | | | | |
| | Mnist | Fashion | Smiles | Eyebrows | Eyebags | Mnist 1% | 5% | Fashion 1% | 5% | Smiles 5% | 10% | Eyebrows 5% | 10% | Eyebags 5% | 10% |
| --- | --- | --- | --- | --- | --- | --- | --- | --- | --- | --- | --- | --- | --- | --- | --- |
| Vanilla IG | 0.998 | 0.759 | 0.354 | 0.155 | 0.143 | 0.469 | 0.508 | 0.109 | 0.196 | 0.076 | 0.085 | 0.097 | 0.104 | 0.117 | 0.131 |
| + (B+W) | **0.999** | 0.901 | 0.584 | 0.422 | 0.361 | 0.379 | 0.631 | 0.083 | 0.217 | 0.149 | 0.185 | 0.209 | 0.233 | 0.146 | 0.195 |
| + Counterfactual | **0.999** | 0.909 | 0.600 | 0.396 | 0.325 | **0.751** | **0.872** | **0.217** | **0.431** | 0.176 | 0.215 | 0.213 | 0.244 | 0.153 | 0.179 |
| Blur IG | 0.973 | 0.818 | 0.368 | 0.144 | 0.136 | 0.017 | 0.102 | 0.016 | 0.076 | 0.015 | 0.019 | 0.014 | 0.017 | 0.008 | 0.009 |
| Guided IG | 0.996 | 0.655 | 0.333 | 0.134 | 0.119 | 0.222 | 0.291 | 0.009 | 0.035 | 0.014 | 0.017 | 0.016 | 0.023 | 0.009 | 0.012 |
| + (B+W) | 0.997 | 0.735 | 0.318 | 0.151 | 0.130 | 0.115 | 0.283 | 0.006 | 0.036 | 0.017 | 0.018 | 0.035 | 0.046 | 0.008 | 0.013 |
| + Counterfactual | **0.999** | 0.879 | 0.360 | 0.277 | 0.206 | 0.715 | 0.833 | 0.168 | 0.326 | 0.056 | 0.063 | 0.137 | 0.152 | 0.062 | 0.081 |
| Generative IG | **0.999** | **0.920** | **0.737** | **0.429** | **0.433** | 0.747 | 0.866 | 0.201 | 0.386 | **0.318** | **0.389** | **0.243** | **0.278** | **0.173** | **0.233** |
| LIME | 0.993 | 0.630 | 0.231 | 0.088 | 0.140 | 0.000 | 0.021 | 0.001 | 0.011 | 0.011 | 0.015 | 0.009 | 0.016 | 0.009 | 0.019 |
| SHAP | 0.994 | 0.900 | | | | 0.119 | 0.319 | 0.080 | 0.222 | | | | | | |
| + Counterfactual | 0.985 | 0.839 | | | | 0.515 | 0.683 | 0.165 | 0.302 | | | | | | |
| CLUE | 0.969 | 0.659 | 0.349 | 0.177 | 0.135 | 0.264 | 0.289 | 0.042 | 0.076 | 0.028 | 0.031 | 0.043 | 0.050 | 0.007 | 0.010 |
| XRAI + IG | 0.991 | 0.750 | 0.541 | 0.230 | 0.156 | 0.023 | 0.093 | 0.010 | 0.037 | 0.053 | 0.101 | 0.036 | 0.056 | 0.018 | 0.028 |
| + (B+W) | 0.992 | 0.811 | 0.637 | 0.312 | 0.236 | 0.002 | 0.035 | 0.009 | 0.044 | 0.121 | 0.206 | 0.067 | 0.103 | 0.028 | 0.057 |
| + Counterfactual | 0.952 | 0.648 | 0.267 | 0.235 | 0.243 | 0.248 | 0.425 | 0.057 | 0.148 | 0.098 | 0.144 | 0.134 | 0.227 | 0.102 | 0.183 |
| XRAI + GIG | 0.990 | 0.671 | 0.173 | 0.098 | 0.054 | 0.012 | 0.054 | 0.003 | 0.016 | 0.019 | 0.030 | 0.006 | 0.012 | 0.003 | 0.005 |
| + (B+W) | 0.988 | 0.699 | 0.118 | 0.120 | 0.043 | 0.001 | 0.018 | 0.002 | 0.012 | 0.021 | 0.032 | 0.016 | 0.027 | 0.002 | 0.004 |
| + Counterfactual | 0.960 | 0.622 | 0.094 | 0.222 | 0.115 | 0.202 | 0.391 | 0.028 | 0.087 | 0.012 | 0.013 | 0.082 | 0.107 | 0.010 | 0.019 |
| XRAI + Gen IG | 0.971 | 0.710 | 0.512 | 0.240 | 0.275 | 0.245 | 0.415 | 0.047 | 0.129 | 0.179 | 0.243 | 0.141 | 0.224 | 0.113 | 0.190 |

In all cases, segmentation-based interpretability methods such as Xrai or *LIME* offer comparatively worse performance. This is due to the complexity associated with segmentation tasks in the data sets selected for this evaluation.

**Blurring setting**. Evaluations are notoriously dependent on the standard deviation setting of the Gaussian kernel. High standard deviation settings lead to blurred images that are significantly out of distribution. This degrades the projected

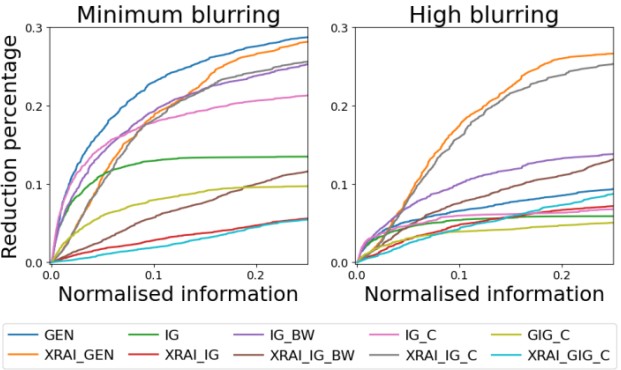

Figure 6: Uncertainty reduction curves for best performing attribution methods on *bags under the eyes*, CelebA data. Left, blurring is set to the minimum feasible value. Right, we assign an arbitrarily large standard deviation.

performance across all attribution methods, as observed in the URC curves displayed in Figure 6, corresponding to the classification model for *bags under the eyes* on CelebA data. Thus, results in Table 1 represent *best* measured performances. Also, we note that attributions produced in combination with Xrai [Kapishnikov et al., 2019] remain consistent across evaluations, a benefit from pre-processing and pixel segmentation leading to highly clustered importances.

**Autoencoder Settings.** The performance of our proposed method plateaus after a certain dimensionality is reached in the latent space representation. Further increasing the complexity of the autoencoder, or changing its training scheme, leads to consistent results. This is a consequence of regularisation terms imposed over optimisation tasks in (2). We note that fiducial points and integration paths are forced to lie in distribution, even within large and overparametrised encoding spaces. A robustness assessment with performance metrics can be found in Section 2 within the supplementary material.

## 4.3    QUALITATIVE EVALUATION

In Figure 7 we find sample uncertainty attribution masks associated with best performing methods, and we offer further examples in Section 3 in the supplementary material. In the figure, attribution masks for vanilla IG and guided

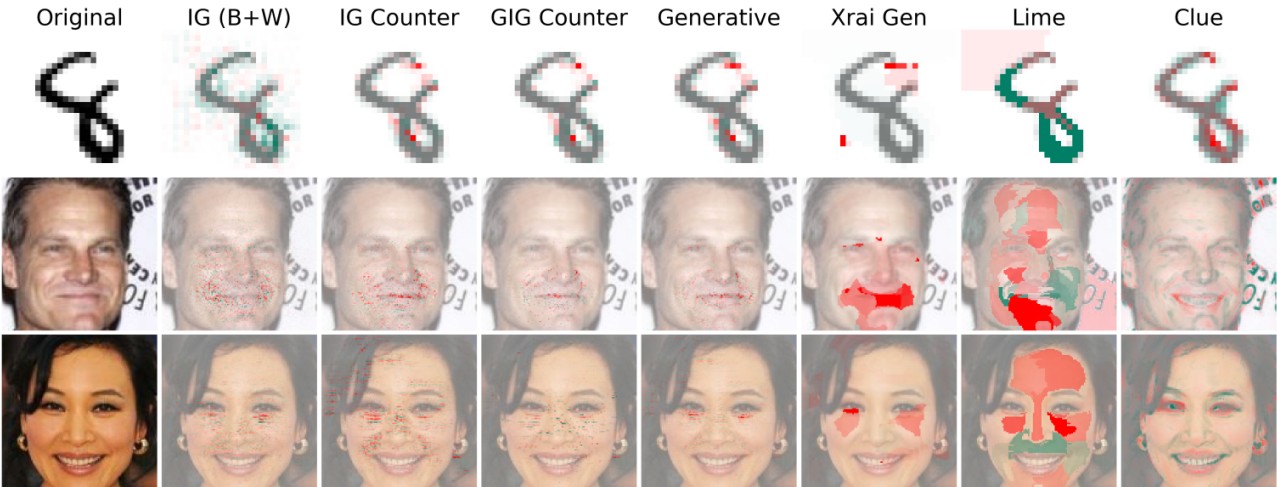

Figure 7: Sample uncertainty attribution masks for selected attribution methods. Masks correspond to digits (top), smiles (mid) and bags under the eyes (bottom).

IG are presented with counterfactual fiducial baselines, in order to avoid noisy saliency masks, such as previously observed in Figure 4. Counterfactual baselines allow to isolate small subsets of pixels that are associated with predictive uncertainty, and producing them requires an autoencoder. In combination with an integration path further defined by a generative model, the attribution method we have presented produces clustered attributions which are de-correlated from raw pixel-value differences between an image and its counterfactual, unlike *Clue* importances. This offers increasingly sparse and easily interpretable uncertainty attributions, which is reportedly associated with better performance in quantitative evaluations [cf. Kapishnikov et al., 2021]. Finally, segmentation based mechanisms do not perform well in the data sets that we have explored, since they do not contain varied objects and items that can be easily segregated.

## 5 DISCUSSION

In this paper, we have introduced a novel framework for the attribution of predictive uncertainties in classification models, which combines path methods, counterfactual explanations and generative models. This is thus an additional tool contributing to improved transparency and interpretability in deep learning applications.

We have further offered comprehensive benchmarks on the multiple approaches for explaining predictive uncertainties, as well as vanilla adaptations of popular score attribution methods. For this purpose, we have leveraged standard feature removal and addition techniques. Our experimental results show that a combination of counterfactual fiducials along with straight or guided path integrals is sufficient to attain best performance in simple classification tasks with greyscale images. However, complex images benefit

from subtle definitions of integration paths that can only be defined through a generative process as described in this paper.

The method presented in this paper is applicable to classification models for data sets where we may feasibly synthesise realistic images through a generative model. This currently includes a variety of application domains, such as human faces, postures, pets, handwriting, clothes, or landscapes [Creswell et al., 2018]. Yet, the scope and ability of such models to synthesise new types of figures is quickly increasing. Also, we evidenced that we do not require a particularly accurate generative process within our method, i.e. the uncertainty attribution procedure we have presented yields top performing results even in the presence of errors and dissimilarities during image reconstructions.

### Author Contributions

I. Perez and P. Skalski contributed equally to this paper.

### Acknowledgements

We thank K. Wong and M. Barsacchi for the support and discussions that helped shape this manuscript. We thank Featurespace for the resources provided during the completion of this research.

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
