# OpenReview forum: "Attribution of Predictive Uncertainties in Classification Models"
_auai.org/UAI/2022/Conference — UAI 2022 Poster_

### Official Review · Reviewer_44M4 · 2022-04-12

**Q2(1) Originality/Novelty:** 2
**Q2(2) Significance/Impact:** 2
**Q2(3) Correctness/Technical Quality:** 2
**Q2(6) Clarity Of Writing:** 3
**Q6 Overall Score:** 6
**Q8 Confidence In Your Score:** 4

**Q1 Summary And Contributions:**

This work investigates an important research question. The proposed approach is well formulated and the experimental results demonstrate the effectiveness of the proposed approach.

**Q2 Assessment Of The Paper:**

More detailed information regarding each of these aspects is given below:

**Q2(4) Quality Of Experiments (Optional):**

2: Fair: The experimental evaluation is weak: important baselines are missing, or the results do not adequately support the main claims.

**Q2(5) Reproducibility:**

2: Fair: Key resources (e.g., proofs, code, data) are unavailable but key details (e.g., proof sketches, experimental setup) are sufficiently well-described for an expert to confidently reproduce the main results.

**Q3 Main Strengths:**

1.	This work investigates an important research problem.
2.	The proposed approach is the combination of several existing ideas and is verified by the experiments.


**Q4 Main Weakness:**

Although this paper performs a few experiments, the scale of the dataset is not large enough. Moreover, the dataset is comparatively simple and more challenging dataset is desired to verify the model performance.

**Q5 Detailed Comments To The Authors:**

This paper is well formulated and is clearly written. Experiments partially verify the effectiveness of the proposed approach. However, a more challenging dataset is desired to valuate the model performance.

**Q7 Justification For Your Score:**

This paper is well formulated and is clearly written. Experiments partially verify the effectiveness of the proposed approach.

**Q9 Complying With Reviewing Instructions:**

1: Yes.

---

### Official Review · Reviewer_RxoL · 2022-04-13

**Q2(1) Originality/Novelty:** 3
**Q2(2) Significance/Impact:** 3
**Q2(3) Correctness/Technical Quality:** 3
**Q2(6) Clarity Of Writing:** 4
**Q6 Overall Score:** 8
**Q8 Confidence In Your Score:** 3

**Q1 Summary And Contributions:**

This paper proposes a novel framework to attribute predictive uncertainties to the input features of a deep-learning model, which combines path integrals, counterfactual explanations, and generative models.

**Q2 Assessment Of The Paper:**

More detailed information regarding each of these aspects is given below:

**Q2(4) Quality Of Experiments (Optional):**

3: Good: The experimental evaluation is adequate, and the results convincingly support the main claims.

**Q2(5) Reproducibility:**

2: Fair: Key resources (e.g., proofs, code, data) are unavailable but key details (e.g., proof sketches, experimental setup) are sufficiently well-described for an expert to confidently reproduce the main results.

**Q3 Main Strengths:**

Solid work, with convincing theory and compelling empirical validation

**Q4 Main Weakness:**

Originality of the work consist essentially of combining previously existing ideas.

**Q5 Detailed Comments To The Authors:**

I'm not particularly knowledgeable about image processing, so I don't have detailed comments to offer. I did not take the appendices into account, but even without them the theory and empirical validation look compelling enough. Nice contribution!

**Q7 Justification For Your Score:**

This is solid work, fully relevant to the main topic of this conference.

**Q9 Complying With Reviewing Instructions:**

1: Yes.

---

### Official Review · Reviewer_BmiF · 2022-04-13

**Q2(1) Originality/Novelty:** 2
**Q2(2) Significance/Impact:** 2
**Q2(3) Correctness/Technical Quality:** 3
**Q2(6) Clarity Of Writing:** 4
**Q6 Overall Score:** 6
**Q8 Confidence In Your Score:** 2

**Q1 Summary And Contributions:**

The paper proposes an approach for attributing predictive
uncertainties in machine learning classifiers. The proposed approach
that combines path integrals, counterfactual explanations and
generative models.

**Q10 Ethical Concerns (Optional):**

No ethical concerns.

**Q2 Assessment Of The Paper:**

More detailed information regarding each of these aspects is given below:

**Q2(4) Quality Of Experiments (Optional):**

3: Good: The experimental evaluation is adequate, and the results convincingly support the main claims.

**Q2(5) Reproducibility:**

3: Good: Key resources (e.g., proofs, code, data) are available and key details (e.g., proofs, experimental setup) are sufficiently well-described for competent researchers to confidently reproduce the main results.

**Q3 Main Strengths:**

The paper covers the topic of predictive uncertainties. The paper is
well-written, and contributes a tool to a concrete problem.

**Q4 Main Weakness:**

The work is heuristic in nature, and so lacking on the theoretical side.

**Q5 Detailed Comments To The Authors:**

The paper reads well.

Some results in Table 1 seem to be of extremely poor quality,
e.g. LIME and some of the XRAI variants. It would be helpful to
explain clearly why that is the case.

**Q7 Justification For Your Score:**

The paper addresses of topic of interest, proposes a solution, and it
is well written.

**Q9 Complying With Reviewing Instructions:**

1: Yes.

---

### Decision · Program_Chairs · 2022-05-15

**Decision:**

Accept (Poster)

**Comment:**

Meta Review: The manuscript introduces a novel framework for the attribution of predictive uncertainties in classification models, which combines path methods, counterfactual explanations (not in a causal sense) and generative models.

While the work does not contain a theoretical analysis of the proposed method, the reviewers have agreed that the proposed methodology is well motivated and shows compelling empirical results.